# Clinical Role of Extraoral Bitter Taste Receptors

**DOI:** 10.3390/ijms21145156

**Published:** 2020-07-21

**Authors:** Joanna Jeruzal-Świątecka, Wojciech Fendler, Wioletta Pietruszewska

**Affiliations:** 1Department of Otolaryngology, Head and Neck Oncology, Medical University of Lodz, 90-419 Lodz, Poland; wioletta.pietruszewska@umed.lodz.pl; 2Department of Biostatistics and Translational Medicine, Medical University of Lodz, 90-419 Lodz, Poland; wojciech.fendler@umed.lodz.pl; 3Department of Radiation Oncology, Dana-Farber Cancer Institute, Boston, MA 02115, USA

**Keywords:** bitter taste receptors, *TAS2R*, biomarker, human tissue, taste, genetic background, receptor signalling, chronic rhinosinusitis, asthma, obesity, diabetes, cancer

## Abstract

Humans can recognise five basic tastes: sweet, sour, salty, bitter and umami. Sour and salty substances are linked to ion channels, while sweet, bitter and umami flavours are transmitted through receptors linked to the G protein (G protein-coupled receptors; GPCRs). There are two main types of GPCRs that transmit information about sweet, umami and bitter tastes—the Tas1r and *TAS2R* families. There are about 25 functional *TAS2R* genes coding bitter taste receptor proteins. They are found not only in the mouth and throat, but also in the intestines, brain, bladder and lower and upper respiratory tract. The determination of their purpose in these locations has become an inspiration for much research. Their presence has also been confirmed in breast cancer cells, ovarian cancer cells and neuroblastoma, revealing a promising new oncological marker. Polymorphisms of *TAS2R38* have been proven to have an influence on the course of chronic rhinosinusitis and upper airway defensive mechanisms. *TAS2R* receptors mediate the bronchodilatory effect in human airway smooth muscle, which may lead to the creation of another medicine group used in asthma or chronic obstructive pulmonary disease. The discovery that functionally compromised *TAS2R* receptors negatively impact glucose homeostasis has produced a new area of diabetes research. In this article, we would like to focus on what facts have been already established in the matter of extraoral *TAS2R* receptors in humans.

## 1. Introduction

Humans can recognise five basic tastes: sweet, sour, salty, bitter and umami, which is the taste of glutamate acid, described as the taste of “broth” or “meat”. Taste is one of the basic senses available to organisms and is used mainly for the chemical analysis of food composition. Taste recognition begins at the taste receptors, which are found in the taste buds on the mucous of the tongue, palate and throat. These taste buds contain elongated receptor cells, which contact the microvilli with substances dissolved in oral cavity fluids. Each specific receptor cell detects a certain type of taste and triggers the potential for transmitting taste information in various mechanisms. Sour and salty substances are linked to ion channels, while sweet, bitter and umami flavours are transmitted through receptors linked to the G protein [1,2,3,4,5,6]. The molecular sensors for all five taste groups act through second messenger systems to initiate neural signalling. These second messenger systems include phospholipase C-, cyclic AMP- and IP_3_-responsive mechanisms. In many cases, the second messenger systems cause the release of calcium from intracellular stores, leading to the initiation of cell depolarisation and the transmission of neural impulses to the brain via the chorda tympani and glossopharyngeal nerves [7].

However, each person experiences flavours a little differently and reacts differently to certain substances. An example of such a reaction may be the recognition of the taste of phenylthiocarbamide (PTC). A chance discovery made by Fox in 1931 revealed that although many individuals perceive PTC as intensely bitter, this substance is relatively tasteless to a large fraction of the population [8]. This discovery led to further research into the genetic determinants of taste sensation. Among other things, it was found that there are two main types of receptors linked with the G protein (G protein-coupled receptors; GPCR_S_) that transmit information about sweet, umami and bitter tastes. They were named the Tas1r family and *TAS2R* family of receptors and the DNA coding regions were designated as the *TAS1R* and *TAS2R* genes. *TAS2R* receptors detect ingested bitter compounds such as toxic plant alkaloids, while TAS1R detects sugars such as glucose and sucrose. There are about 25 functional *TAS2R* genes that code bitter taste receptor proteins, while there are only three *TAS1R* genes that code sweet and umami taste receptors [2,9] (Figure 1). It has been proven that these taste receptors are found not only in the mouth and throat, but also in the intestines, brain, bladder, lower and upper respiratory tract and pancreas [10] (Table 1). Research has therefore been undertaken to determine the purpose of their presence in these locations, and the suspicion has been raised that they must be involved in initiating the body’s immune system reactions as an additional defence mechanism. Much research has been already undertaken in mouse and rat models, and this work has been reviewed in other articles [11,12]. In this article, we would like to focus on what facts have been already established in the matter of extraoral *TAS2R* receptors in humans.

## 2. *TAS2R* Bitter Taste Receptor Activation and Intracellular Signalling Cascades 

*TAS2Rs* are, as mentioned before, G protein-like receptors. GPCRs specifically connected to sweet and bitter taste receptors couple to specific intracellular G proteins. G α-gustducin is involved in sweet and bitter taste transduction, and G γ13 is exclusively for bitter taste perception [39,40,41]. When *TAS2R* receptors are stimulated with known agonists such as denatonium, 6-n-propyl-2-thiouracil (PROP) or phenylthiocarbamide (PTC), the reaction goes in two ways. Activated α-gustducin stimulates a taste phosphodiesterase (PDE) to hydrolyse cAMP. These decreased levels of cAMP may disinhibit cyclic nucleotide-inhibited channels (cNMP) and in consequence, elevate intracellular Ca^2+^, promoting neurotransmitter release [42]. The subsequent steps in the α-gustducin-PDE-cNMP pathway are still uncertain. The second pathway leads through the β and γ subunits of gustducin (identified as Gβ3 and Gγ13) which activate phospholipase C isoform β2 (PLCβ2) to generate inositol trisphosphate (IP_3_). The released IP_3_ binds to IP3 receptor type III (IP3RIII). This mechanism leads to the release of Ca^2+^ from internal stores [43]. Elevated levels of Ca^2+^ activate transient receptor potential proteins (TRP) within the taste receptor, especially TRPM5, which is known to be abundantly expressed in taste receptor cells that are involved in sweet, amino acid and bitter perception [44,45,46]. TRPM5 is a necessary component of sweet, amino acid and bitter taste reception, although it is still not certain what the true role of Ca^2+^ is in its response [47].

A fascinating discovery is also that the number of compounds perceived by humans as bitter is much larger than the number of human *TAS2R* genes [48]. This implies that each *TAS2R* receptor responds to more than one bitter ligand [49]. Several *TAS2Rs*, such as *TAS2R7*, *TAS2R14* and *TAS2R46* are broadly tuned to detect stimuli of different chemical classes, while others, such as *TAS2R4*, *TAS2R5*, *TAS2R8* and *TAS2R43*, appear to be more specific and activated by one or a few agonists [50,51,52]. It appears that different *TAS2R* alleles may have different profiles of ligand specificity [53,54,55]. The number of *TAS2R* genes may not, therefore, be a spectrum-limiting element for the receptors, which may include many variants each leading to a different ligand preference.

## 3. The Role of Bitter Taste Receptors in Rhinosinusitis

Recent data demonstrate that chronic rhinosinusitis (CRS) affects approximately 5–16% of the general population both in Europe [56] and the USA [57]. It is divided into two specific disease entities: chronic rhinosinusitis with (CRSwNP) and without nasal polyps (CRSsNP), and it depends on the presence of polypous lesions in the nasal cavities and/or paranasal sinuses. There are many factors attributed with a role in the etiopathogenesis of CRS, including allergy, congenital immunodeficiencies, ciliary impairment as in patients with Kartagener’s syndrome, primary ciliary dyskinesia and cystic fibrosis, bacteria forming biofilms, environmental factors and genetic factors [58]. One of the genetic factors is the type of bitter taste receptor in the upper airway. The ability to regulate congenital immunity, especially in the respiratory system, is increasingly attributed to bitter taste receptors (*TAS2Rs*), especially *TAS2R38* [59,60].

When epithelial cells are stimulated with known agonists of *TAS2R38* such as phenylthiocarbamide (PTC), they exhibit low-level calcium responses that activate nitric oxide (NO) synthase (NOS) and lead to robust intracellular NO production [61]. In 2017, Yan et al. confirmed the same signalling pattern for *TAS2R4* and *TAS2R16* receptors present in sinonasal tissue [62]. This signalling pathway included two of the important components of the well-established taste signal transduction cascade, namely phospholipase C isoform β2 (PLCβ2) and the transient receptor potential cation channel subfamily M member 5 (TRPM5) ion channel (Figure 2).

Nitric oxide and its derivatives are damaging to bacterial membranes, enzymes and DNA. They also increase ciliary beat frequency through the activation of guanylyl cyclase and protein kinase G, which phosphorylate ciliary proteins that accelerate mucociliary clearance. The identification of physiologic ligands that activate *TAS2R38* is further evidence in support of the role of *TAS2R38* in airway immunity. Two major *P. aeruginosa* acyl-homoserine lactones (AHLs), *N*-butyryl-l-homoserine lactone (C4HSL) and *N*-3-oxo-dodecanoyl-l-homoserine lactone (C12HSL) [63,64], were identified as agonists of *TAS2R38* [19]. Through studies with purified AHLs and conditioned medium from wild-type *P. aeruginosa*, it was demonstrated that *TAS2R38* detects physiological concentrations of AHLs, which results in the activation of calcium-dependent NO production [65]. Many Gram negative species secrete AHLs and therefore, *TAS2R38* is likely to function in airway ciliated cells as a sentinel receptor for detecting invading Gram negative bacteria, triggering a critical defensive bactericidal response. 

Patients with the same risk factors of a similar age and of the same sex may have a completely different course of the disease. For years, clinicians have been looking for factors influencing the course of the disease and treatment in order to be able to propose treatment that is best suited to each patient. It was established that single nucleotide polymorphisms (SNPs) in the *TAS2R38* gene may contribute to individual differences in susceptibility to respiratory infections, in particular, to chronic rhinosinusitis (CRS) [55,66]. The efficacy of this inborn immune defence within the nasal and sinus mucosa differs depending on the three most common polymorphisms in the *TAS2R38* gene. The variants are associated with amino acid residues at positions 49 (rs713598 C/G p.Ala49Pro), 262 (rs1726866 G/. p.Ala262Val) and 296 (rs10246939 T/C p.Ile296Val). These three polymorphisms are connected and generate two common haplotypes. In the “protective” type, the *TAS2R38* allele codes proline, alanine and valine (PAV), and in the “non-protective” type, the receptor allele codes alanine, valine and isoleucine (AVI), respectively. Patients who are homozygous PAV/PAV are considered “supertasters”, AVI/AVI are “nontasters”, and the heterozygous PAV/AVI show the widest range of bitter taste reception [1]. Other haplotypes, namely AVV, AAI, PAI and PVI, have been observed rarely (1–5%) or in specific populations [67]. Many recently conducted studies show that people affected by chronic rhinosinusitis are much less likely to be PAV/PAV homozygotes [68,69] (Table 2). Adappa et al. presented a pilot study in 2013 where they hypothesised that the PAV/PAV genotype would be less likely to occur in CRS patients who have failed medical therapy and require primary functional endoscopic sinus surgery (FESS) and that even after surgery these patients would have clinically improved outcomes in comparison with patients with PAV/AVI or AVI/AVI genotypes. Due to the small number of supertasters in this cohort, no conclusions could be reached but these initial suggestions led to further research [70]. This was followed by a subsequent prospective study in a bigger patient group (N = 70) undergoing primary FESS. The distribution of diplotypes in the CRS patients was 37% AVI/AVI, 54% AVI/PAV and 8.5% PAV/PAV, which was significantly different from the distributions found in the general Philadelphia population. No significant differences were found in the allele distribution with respect to other risk factors, such as asthma, allergies, aspirin sensitivity, diabetes, smoking exposure or nasal polyposis, suggesting that *TAS2R38* is an independent risk factor for CRS requiring FESS [71]. In 2016 the same authors defined the prognostic value of the PAV/PAV genotype as favourable in terms of outcome after surgery (QoL) in CRSsNS patients [72]. Dżaman et al. found *TAS2R38* to be highly expressed in the sinonasal mucosa of CRS patients. The heterozygote frequency (AVI/PAV) was the highest and the protective genotype (PAV/PAV) was noticed in the lowest frequency and connected with a lower average value of CT score compared to AVI/AVI genotypes (*p* = 0.01) [73]. Cantone et al. confirmed that the nonfunctional genotype is more frequent among CRS patients than in the general population (25% vs.18.4%), that airway Gram negative infections are primarily associated with AVI and that biofilm formation is prevalent in CRS patients with the AVI nontaster phenotype, which confirmed an inverse correlation between *TAS2R38* functionality and Gram negative infections in patients with CRSwNP [74,75]. Cohen et al. demonstrated that upper airway epithelial cells from individuals with one or two nonfunctional *TAS2R38* alleles (AVI) have significantly blunted nitric oxide and ciliary responses following exposure to Gram negative quorum-sensing molecules and that these individuals are more likely to be infected with Gram negative bacteria such as *Pseudomonas aeruginosa* than those with two functional receptor alleles, and to develop CRS [76]. In contrary, an Italian research team published results questioning this theory. Gallo et al. reported no significant difference in the distribution of the three more common genotypes (PAV/PAV, PAV/AVI, AVI/AVI) by comparing CRS patients and healthy controls. They also found no significant differences in genotype distribution between CRSwNP and CRSsNP, nor any significant association between any CRS-related factors and a particular genotype in their study group [77].

Some new facts have been presented recently by Freund et al. and Hariri et al. on the role of *TAS2Rs* other than *TAS2R38* in rhinosinusitis [33,34]. The presence of *TAS2R4*, *TAS2R14* and *TAS2R16* receptors was confirmed in the sinonasal cilia, and they were found to be activated by the Pseudomonas quinolone signal (2-heptyl-3-hydroxy-4-quinolone) and 4-hydroxy-2-heptyl quinolone.

The incipient role of *TAS2R38* and its polymorphisms in chronic rhinosinusitis inspired a study involving patients with cystic fibrosis (CF) homozygous for the pathogenic ΔF508 mutation. Adappa et al. examined a group of 69 patients with a *TAS2R38* genotype of PAV/PAV, PAV/AVI, or AVI/AVI and homozygous for the ΔF508 CF mutation [78]. Out of this group, they selected 49 patients aged 18–32 years. Subsequently, it was found that CF patients with *TAS2R38* functional alleles (PAV/PAV) had better rhinologic quality of life scores (SNOT-22) compared to CF patients with non-functional PAV/AVI and AVI/AVI genotypes. They also found rhinologic symptoms to be significantly lower in PAV/PAV patients than non-PAV/PAV patients. It can be speculated that the use of *TAS2R38* as a genetic marker in CF patients offers a novel approach to evaluating sinonasal disease in this population and may ultimately help in future treatment modalities.

## 4. *TAS2R* Receptors in Lower Airways

Studies on the primary cilia of the lower respiratory tract and their numerous functions have also led to the discovery of taste receptors in their structure [79]. In 2009, Shah et al. hypothesised that motile cilia might sense noxious stimuli [15]. They examined airway epithelial samples isolated from eight human donors and proved that only ciliated epithelial cells expressed *TAS2R* receptors (*TAS2R4, TAS2R38*, *TAS2R43* and *TAS2R46*). They were localised specifically in the cilia, which was done by acetylated α-tubulin immunostaining. To determine whether the *TAS2R* signalling pathway was functional, Shah et al. applied bitter compounds (denatonium, thujone, salicin, quinine and nicotine) to differentiated human airway epithelia and measured changes in intracellular Ca^2+^ [Ca^2+^]_i_ concentrations. They all induced transient, dose-dependent increases in intracellular Ca^2+^. Researchers have also determined the correlation between [Ca^2+^]_i_ increase and increased ciliary beat frequency of ~25%, and that this information can rapidly spread through gap junctions to adjacent nonciliated cells. 

Deshpande et al. studied the *TAS2R* receptors in cultured human airway smooth muscle (ASM) and determined their potential role. They performed quantitative RT-PCR studies with primers for 25 *TAS2R* genes and discovered that *TAS2R10*, *TAS2R14* and *TAS2R31* were the most highly expressed in human ASM [16]. This discovery was confirmed by Jaggupilli et al. By studying cystic fibrosis bronchial epithelial cells (CuFi-1), normal bronchial epithelial cells (NuLi-1) and ASM cells, they confirmed the expression of *TAS2Rs* (14, 20 > 3, 4, 5, 50 > 10, 13, 19) in CuFi-1 and NuLi-1. They also demonstrated that ASM almost exclusively expressed *TAS2R10*, *TAS2R14* and *TAS2R31* receptors [30]. A study by Deshpande et al. also linked the expression of bitter receptors to bitter tastant-mediated [Ca^2+^]_i_ signalling. The response to the increase in [Ca^2+^]_i_ was ablated by the Gβγ inhibitor gallein and the PLCβ inhibitor U73122, and partially inhibited by the inositol-3-phosphate (IP_3_) receptor antagonist 2APB. However, the exact bronchodilatation mechanism itself is still not entirely certain. Deshpande et al. suggested that bitter tastants activate the canonical bitter taste signalling pathway (i.e., *TAS2R*-gustducin-phospholipase (PLCβ2)- inositol 1,4,5-triphosphate receptor [IP_3_R]) to increase focal Ca^2+^ release from the endoplasmic reticulum, which then activate large-conductance Ca^2+^-activated K^+^ channels, thereby hyperpolarising the membrane [16]. However, Zhang et al. subsequently demonstrated through patch-clamp recordings that bitter tastants do not activate large-conductance Ca^2+-^-activated K^+^ channels, rather they inhibit them [80]. However, it should be taken into account that while both research teams used chloroquine as the main bitter tastant, they performed tests on different types of tissues. Deshpande used cultured human ASM, and Zhang used mouse ASM cells. 

These differences in outcomes inspired Zhang’s research team to seek the ultimate answer by studying this process in freshly isolated ASM tissues and cells [81]. They revealed that bitter tastants reverse the increase in [Ca^2+^]_i_ evoked by bronchoconstrictors, leading to bronchodilation. This reversal is mediated by the suppression of l- type voltage-dependent calcium channels (VDCCs) in a gustducin β and γ subunit-dependent, yet PLCβ2- and IP_3_R-independent manner. It seems that *TAS2R* activation in ASM stimulates two opposing Ca^2+^ signalling pathways, both mediated by the Gβ and γ subunits, which increases [Ca^2+^]_i_ at rest, but blocks activated l- type VDCCs, reversing the contraction they cause. However, Grassin-Delyle et al., by studying human lung tissue obtained from macroscopically healthy parts of the lungs from 77 patients, established that none of the signalling pathways targeted by current bronchodilators, nor the inhibition of BKCa or L-type voltage-gated calcium channels, could fully explain the *TAS2R* agonist-induced relaxation of human isolated bronchi [20]. Evidently, the exact mechanism has not yet been precisely defined and requires further study, but the bronchodilatation potential of *TAS2Rs* cannot be underestimated. What is certain is that bitter tastants may represent a new class of compounds with potential as potent bronchodilators, and that their potential should be further exanimated. 

These discoveries led to the initiation of research into the role of *TAS2Rs* in inflammatory processes in the lower respiratory tract, such as asthma or chronic obstructive pulmonary disease (COPD). Asthma is a chronic inflammatory disorder of the airways associated with increased airway hyperresponsiveness leading to recurrent episodes of wheezing, breathlessness, chest tightness and coughing [82]. The chronic airway inflammation of asthma is characterised by airway wall infiltration by T-lymphocytes of the T-helper (Th) type 2 phenotype, eosinophils, macrophages/monocytes and mast cells. COPD is a chronic inflammatory disease of the lower airways characterised by the destruction of the lung parenchyma (pulmonary emphysema), the inflammation of the peripheral airways (respiratory bronchiolitis) and the inflammation of the central airways. Dysfunction of ASM cells, a major cell type in the respiratory tree, plays a pivotal role in promoting the progression of these diseases and in contributing to the symptoms of these diseases. With their ability to contract and relax, these cells regulate the diameter and length of conducting airways, controlling dead space and resistance to airflow to and from gas-exchanging areas [83,84].

In 2013, Orsmark-Pietras et al. established that *TAS2R* receptors can be found on isolated leucocytes in a group of asthmatic and healthy children. The expression of all *TAS2Rs* was significantly greater in the lymphocyte population compared to monocytes in both groups [21]. The expression of most *TAS2Rs* was higher in children with severe asthma compared to the healthy controls, especially *TAS2R13*, *TAS2R14* and *TAS2R19*. In asthmatic children, *TAS2R5* expression in mixed-blood leukocytes also showed a statistically significant correlation with bronchial hyperresponsiveness to methacholine. Moreover, two of the bitter tastants, chloroquine and denatonium, were found to significantly inhibit the release of TNFα, IL-13 and MCP-1. Chloroquine also inhibited the release of IL-1β, IL-2, IL-4, IL-5, IL-10, IL-17, G-CSF, GM-CSF, IFNγ and PGE2 in a dose-dependent manner. The presence of *TAS2R* receptors on monocytes and their functionality was confirmed by Gopallawa et al. [85].

On the other hand, Robinett et al. investigated *TAS2R*-mediated bronchodilatory effects in human ASM cells derived from asthmatic patients and controls. *TAS2R* expression and *TAS2R*-mediated bronchodilatory effects were found to be independent of the disease state and IL-13-induced inflammatory environment [86]. Sharma et al. investigated yet another important aspect of chronic inflammatory pulmonary disease: airway remodelling, which is a hallmark feature of asthma and COPD [87]. The findings from this study demonstrated the antimitogenic effect of *TAS2R* agonists. Mast cell infiltration of ASM in asthma is also one of the very important contributions to airway hyperresponsiveness in this medical condition [88]. Ekoff et al. examined the expression of nine *TAS2R* genes in human cord blood-derived mast cells (CBMCs), the mast cell line HMC1.2 and the mast cell line LAD-2 [25]. The first two mast cell lines expressed several *TAS2Rs*. CBMCs expressed *TAS2R4*, *46*, *14*, *19*, *20*, *3*, *10*, *13* and *5* in decreasing order and HMC1.2 expressed *TAS2R4*, *3*, *46*, *14*, *10*, *5*, *19*, *13* and *20*, also in decreasing order. More importantly, *TAS2Rs* mediated significant inhibition of the release of histamine and PGD2 from IgE-receptor-activated primary human mast cells, which suggests that *TAS2R* receptors may mediate anti-inflammatory responses. *TAS2R* expression and the effects of *TAS2R* agonists on lipopolysaccharide (LPS)-induced cytokine release in human lung macrophages (LMs) were studied by Grassin-Delyle et al. [36]. They found that 16 *TAS2Rs* are expressed in human LMs with the most abundant transcripts being of *TAS2R8*, *14*, *19*, *39* and *46*. They established that *TAS2R* agonists inhibit LPS-induced cytokine release by LMs—a process that is not mediated by the release of IL-10.

Deshpande et al. and Zhang et al. suggested that bitter tastants may cause greater ASM relaxation than β2 adrenergic agonists, the most commonly used bronchodilators for treating asthma and COPD [16,81]. However, there are still some doubts concerning this matter [89,90]. What is certain is that bitter tastants represent a new class of compounds with potential as potent bronchodilators, and that their potential should be further exanimated. One of the many challenging aspects in the development of *TAS2Rs* as therapeutic targets is the dearth of receptor subtype-specific agonists with a refined pharmacological profile. Although there are well-characterised bitter compounds at our disposal, agonist-receptor subtype promiscuity has forestalled progression to clinical trials [91,92].

## 5. Bitter Taste Receptors in Other Systems and Diseases 

*TAS2R* receptors were also found in many other internal organs and the skin. A study by Clark et al. indicated that *TAS2Rs* couple the detection of bitter-tasting compounds to changes in thyrocyte function and T3/T4 production [26]. By studying the human thyrocyte line (Nthy Ori 3-1) and 15 *TAS2R* bitter taste receptors, they discovered that *TAS2R* agonists inhibit TSH-dependent Ca^2+^ increases in thyrocytes. They also demonstrated that nonsynonymous coding SNP (rs5020531), located within the *TAS2R42* gene, showed a significant association with elevated fT3 or fT4 levels. This indicates that any difference in the primary structure of a thyroid-expressed *TAS2R* is associated with differences in the circulating levels of thyroid hormones. Interestingly, the human heart also seems to sense bitter compounds through the expression of *TAS2Rs* on myocytes and fibroblasts [22]. *TAS2R14* was found to be highly expressed in those tissues, and the authors suggested that *TAS2Rs* might be involved in adapting the heart’s function in response to the presence of toxic substances in the blood. In addition, coding SNP (rs1376251) in *TAS2R50* is believed to be associated with higher myocardial infarction risk [93,94].

Lund et al. studied the possibilities for human bone marrow stromal-derived cells (MSCs) and vascular smooth muscle cells (VSMCs) to interact with their environment through novel receptors [23]. They found that both MSC and VSMC cells express the bitter taste receptor *TAS2R46* and are fully functional. Another study by Wölfle showed *TAS2R1* and *TAS2R38* expression throughout the epidermis and weak expression in the basal cells [28]. The keratinocyte cell line HaCaT and human primary keratinocytes (HPKs) both expressed *TAS2Rs* and showed Ca^2+^ responses and increases in the expression of differentiation marker genes in response to both diphenidol and amarogentin. In 2018, Shaw et al. also studied the expression of *TAS2Rs* in human skin [35]. They demonstrated differences in the expression of bitter taste receptors according to sex, age and sun exposure. There were significantly lower expression levels in sun-exposed skin for *TAS2R14*, *TAS2R30* and *TAS2R42*, but they were higher for *TAS2R60*. In skin from the suprapubic area, females’ expression was significantly higher for *TAS2R3*, *TAS2R4* and *TAS2R8*. In skin from the lower leg, females’ expression was significantly lower for *TAS2R3*, *TAS2R9* and *TAS2R14*. Additionally, there was a positive correlation between increasing age and the expression of the *TAS2R5* gene, but only in non-sun-exposed skin.

Zheng et al. studied the problem of preterm birth focusing their work on finding new molecular targets that might mediate myometrial relaxation and possibilities for developing new classes of effective and safe tocolytics [32]. Considering that *TAS2R* receptors are an attractive target for smooth muscle relaxants Zheng studied their expression in human myometrium cells obtained from peripartum hysterectomies performed on multiparous reproductive-age women at the time of indicated deliveries for medical reasons (i.e., placenta accreta, postpartum hemorrhage resulting from uterine atony) and cultured hTERT-HM cells. Transcripts for *TAS2R4/5/10/13/14* were detected in 25 freshly collected human myometrial tissues and for *TAS2R3/4/5/7/8/10/13/14/31/39/42/43/45* and *50* in cultured hTERT-HM cells. This study also proved that bitter tastants such as chloroquine and 1,10-phenanthroline almost fully reverse OT- and KCl-induced contraction of the human myometrium. This effect was greater than that of clinically used tocolytics such as MgSO4, indomethacin, nifedipine and albuterol. *TAS2R14* was indicated as the receptor that mediates the chloroquine-induced effect on [Ca^2+^]_i_ in human myometrial cells. Most recently, a study on semen cells and testis tissue was conducted by Governini et al. [38]. Sperm was donated by 15 Caucasian males undergoing fertility examination. Samples of testicular tissue were obtained from six patients undergoing diagnostic testicular biopsy for obstructive azoospermia. Sperm cells were examined before and after in vitro selection and capacitation. The researchers demonstrated that both human testis tissue and semen cells express *TAS2Rs*. Some inter-individual variation for a single *TAS2R* was established. *TAS2R3/4/19* and *TAS2R43* were detectable in about 73% of semen samples (11/15), whereas *TAS2R14* was found in 80% of the probes (12/15). Interestingly, again *TAS2R14* was the most expressed bitter receptor subtype in both testis tissue and sperm cells. The authors also observed that in vitro capacitation significantly affects both the expression and the subcellular localisation of these receptors in isolated spermatozoa.

*TAS2R* expression was reported to be dysregulated and to potentially play a role in side-effects induced by antipsychotic drugs in the cortex of Parkinson’s disease and schizophrenia patients [95,96]. Deckmann et al. reported the presence of solitary chemosensory cells (SCCs) in the human urethra. Their observations suggest that those cells may play a sentinel function as upon the detection of possibly toxic substances, a protective reflex pathway is initiated, resulting in increased urine flow, which might serve to flush pathogens out of the body [97]. Furthermore, *TAS2R38* polymorphism (AVI/AVI) has been associated with the risk of dental caries [98]. Another extraordinary discovery by Wölfle et al. showed strong expression of *TAS2R38* in the amniotic epithelium, syncytiotrophoblast and decidua cells of the human placenta. Diphenidol stimulation of these cells (JEG-3) demonstrated the functionality of the *TAS2Rs* by inducing calcium influx [29]. *TAS2Rs* thus appear to be located in different parts of the human body, and determining their exact significance requires further research.

## 6. The Role of *TAS2R* Receptors in Eating Behaviours, Obesity and Diabetes 

Eating habits and preferences are modified by taste perception. The growing interest in the causes of specific eating behaviours has encouraged research into genetic conditioning. Taste perception differences are formed by variations in receptor function for sweet, fat, and bitter tastes. The consumption of vegetables rich in antioxidants may be lower in individuals with higher sensitivity to bitter tastants [13,99,100,101]. During our lifetime, our food preferences can be modified by socialisation, the environment in which we live and even culture. However, it has been proven that we also inherit some of these preferences [102]. Cont et al. investigated the differences in food intake in infants [37]. They genotyped 131 infants, of whom 45 were bitter-insensitive (genotype AVI/AVI) (34.3%), and 86 (65.6%) were bitter-sensitive. They were divided as follows: 28.2% PAV/PAV (37/131) and 37.4% PAV/AVI (49/131). Infants who were insensitive to bitter taste were more likely to consume the whole first complementary food meal at the first attempt, compared to sensitive ones (either the AVI/PAV or PAV/PAV genotypes). Moreover, bitter-insensitive infants consumed the whole volume of the first complementary food in fewer days than bitter-sensitive ones. It appears that supertasters may also have a corresponding lower intake of high fat and high sugar foods [103]. This may protect the supertasters against becoming overweight or obese [104].

The *TAS2R38* receptor was also linked to the avoidance of alcohol consumption [105] and cigarette smoking [106], which might have an impact on cardiovascular disease [107]. A large observational study in Spain revealed a significant association between obesity and the *TAS2R38* AVI/AVI haplotype in females (N = 1319). These associations were not observed in male participants, and they were more robust in subjects aged <40 years [108]. Those findings were further supported by a study that established a significant association between three *TAS2R38* variants and body fat percentage in females [109]. Super-tasting was also shown to have an impact on body mass index (BMI) by lowering it [110]. A study by Sharma et al. also confirmed that female PTC tasters aged 14–16 years have a higher mean average skinfold, percentage of body fat, fat mass index and fat-free mass index [111]. In the oral cavity, *TAS2Rs* are involved in the perception of bitter tastants, while in the lower gastrointestinal (GI) tract they have roles in chemoreception and the regulation of GI function. 

Ghrelin is a hunger hormone that is mainly produced by the X/A-like cells in rats or in human P/D1-type cells of the gastric oxyntic mucosa, and to a lesser extent in the small intestine. Quinine decreased plasma ghrelin levels in the fasting state [112,113] while glucose inhibited ghrelin secretion [114]. In 2019, Wang et al. studied the impact of obesity on ghrelin secretion and the role of *TAS2R* receptors in this process [115]. The entire stomach and part of the small intestine (duodenum and jejunum) were collected from circulatory death or brain death organ donors (lean). Part of the stomach and a segment of the jejunum were obtained from obese nondiabetic patients who had undergone sleeve gastrectomy or Roux-en-Y gastric bypass surgery. They discovered that *TAS2R4*, *TAS2R10* and *TAS2R3* were expressed in both the stomach and the jejunum, whereas *TAS2R38* was only expressed in the jejunum. Additionally, the mRNA expression of *TAS2R4*, *TAS2R10* and *TAS2R38* was significantly altered by obesity in a tissue-dependent manner. The differences in mRNA expression between patient groups were BMI-dependent and were not related to age or gender. Chloroquine (*TAS2R3/7/10/14/39* agonist), emetine (*TAS2R1* agonist) and 1,10- phenanthroline (*TAS2R5* agonist) also increased ghrelin secretion in a concentration-dependent manner in fundic cultures from obese patients. The effect of denatonium benzoate (DB) on ghrelin secretion mainly involves *TAS2R10*. 

Latorre et al. studied *TAS2R38* expression in the colonic mucosa and its role in overweight/obese (OW/OB) versus lean (NW) patients. It was shown that *TAS2R38* is expressed by the human colonic mucosa, where it is localised to distinct types of enteroendocrine cells (EEC), including cholecystokinin (CCK), and glucagon-like peptide-1 (GLP-1) and peptide YY (PYY) cells [31]. It is upregulated in overweight and obese subjects, as indicated by the marked and significant increase in the number of *TAS2R38*-IR cells without changes in the overall density of EEC cells in the colonic mucosa of OW/OB vs. NW subjects. Agonists of the same *TAS2R*, such as phenylthiocarbamide, were found to increase ATP-binding cassette B1 (ABCB1) expression and increase the efflux activity of ABCB1. This suggests that *TAS2R* signalling limits the absorption of bitter-tasting/toxic substances through the modulation of gut efflux membrane transporters [18].

The expression of transcripts encoding *TAS2R* family members, hT1R3 and G α-gustducin in the human colon and the human intestinal endocrine cell lines (HuTu-80 and NCI-H716 cells) was confirmed by Rozengurt et al. The stimulation of HuTu-80 or NCI-H716 cells with PTC induced a rapid increase in the intracellular Ca^2+^ concentration in these cells [14]. The identification of G α-gustducin and chemosensory receptors that perceive chemical components of ingested substances, including drugs and toxins, in open enteroendocrine L cells has important implications for understanding molecular sensing in the human gastrointestinal tract and for developing novel therapeutic compounds that modify the function of these receptors in the gut. Other *TAS2R* receptor ligands were found to have a role in gastrointestinal regulation. An extract of the plant *Hoodia gordonii* induced the release of CCK from the human enteroendocrine cell line HuTu-80, probably via the activation of the *TAS2R14* receptor [17], while berberine stimulates GLP-1 secretion through the activation of *TAS2Rs* and the taste signalling pathway [27]. The expression of *TAS2Rs*, including cognate *TAS2Rs* for caffeine, was shown in human gastric epithelial cells of the corpus/fundus and in HGT-1 cells. Caffeine, generally known as a stimulant of gastric acid secretion (GAS), was discovered to stimulate proton secretion depending on one of its cognate receptors, *TAS2R43* [116]. All these data have led to further research, including on pathological conditions like diabetes [117]. 

Dotson et al. conducted the Amish Family Diabetes Study. They assessed the association of taste receptor variants with indicators of glucose dysregulation, including a diagnosis of type 2 diabetes mellitus and high levels of blood glucose and insulin during an oral glucose tolerance test [118]. They reported that a *TAS2R* haplotype is associated with altered glucose and insulin homeostasis and that one SNP within this haplotype disrupts the normal responses of a single receptor, *TAS2R9*, to its cognate ligands ofloxacin, procainamide and pirenzapine. This led to the conclusion that a functionally compromised *TAS2R* receptor negatively impacts glucose homeostasis, providing an important link between alimentary chemosensation and metabolic disease. *TAS2R* receptors may be a new target for the treatment of diabetes. It is speculated that the large number of *TAS2Rs* expressed in the intestines, each with different ligand specificities, offers diverse targets that may allow for selectivity and specificity. Thus, exploring whether any known bitter-tasting drugs have positive effects on glucose homeostasis would be of interest [119,120]. So far, however, no drug discovery activities involving TAS1Rs and/or *TAS2Rs* for the potential treatment of type 2 diabetes appear to be underway.

## 7. *TAS2Rs*’ Impact on Longevity

*TAS2R* genes have been widely associated with individual differences in taste perception, food preferences, diet, nutrition, immune responses and pathophysiological mechanisms. Could it be that by regulating food absorption processing and metabolism, they modulate the ageing process? In 2012, Campa et al. investigated the possible associations between longevity and the common genetic variation at the three *TAS2R* clusters on chromosomes 5, 7 and 12 in a population of 941 individuals ranging in age from 20 to 106 years from the south of Italy [121]. They found that five polymorphisms were associated with longevity: three in the *TAS2R16* gene (rs6466849, rs860170 and rs978739), one in the *TAS2R4* gene (rs2233998) and two in the *TAS2R5* (rs2227264) gene, but only rs978739 reached statistical significance. Melis et al. studied the differences in genotype distribution and haplotype frequency of the *TAS2R38* gene in a group of 94 centenarian and near-centenarian inhabitants of Sardinia, Italy, while comparing them to two control cohorts of 18–35 years of age and 36–85 years of age [122]. Interestingly, the results showed an increased frequency of subjects carrying the homozygous genotype for the functional variant of *TAS2R38* (PAV/PAV) and a decreased frequency of those with the homozygous genotype for the non-functional form (AVI/AVI) in the centenarian cohort, compared to those determined in the two control groups. These data suggest that *TAS2R* variants are involved in the molecular physiological mechanisms involved in the biological process of ageing.

## 8. *TAS2Rs* in Cancer

The constant search for new therapies and new possibilities to monitor the course of disease in oncology has directed scientists towards the bitter taste receptors. In 2014, Singh et al. reported a study on three different mammary epithelial cells: the highly metastatic breast cancer cell line MDA-MB-231, the poorly metastatic cell line MCF-7 and the non-cancerous mammary epithelial cell line MCF-10A [24]. They characterised the expression of *TAS2R1*, *TAS2R4*, *TAS2R10*, *TAS2R38* and *TAS2R49* in those cells, finding that *TAS2R4* is expressed at 40–70% in mammary epithelial cells in comparison to the commonly used breast cancer marker proteins, estrogen receptor and E-cadherin. What is more, these endogenous *TAS2Rs* may be functional because an increase in intracellular calcium mobilisation was observed after the application of the bitter agonists quinine, dextromethorphan, and phenylthiocarbamide. In 2017, Jaggupilli et al. showed a significant difference in expression between normal and cancerous mammary cell lines for *TAS2R14* and *TAS2R20(49*) [30]. To determinate the ultimate difference in *TAS2R4* and *TAS2R14* expression in those cells, in 2019, Singh et al. [123] studied the same lines of mammary cells and established that *TAS2R*4 and *TAS2R*14 expression patterns were dissimilar, with decreased levels of *TAS2R4* and increased levels of *TAS2R14* in breast cancer clinical samples compared to non-cancerous controls. The activation of *TAS2Rs* with their respective agonist elicited physiological responses in metastatic breast cancer cells, and no responses were seen in non-tumourigenic breast epithelial cells. Agonist activation of *TAS2Rs* induced anti-proliferative, pro-apoptotic and anti-migratory responses in highly metastatic breast cancer cells. Taken together, these findings demonstrate that the chemosensory *TAS2R* signalling network is involved in evoking physiological responses in the metastatic breast cancer cell line. 

Ovarian and prostate cancers are among the most lethal gynaecological and urological cancers in the world [124]. Martin et al. decided to determine whether bitter taste receptors are expressed in those two types of cancer [125]. They investigated epithelial ovarian cell lines (OVCAR8, IgROV1, SKOV3, OVCAR4) and prostate cell lines (PC3, LNCaP, DU145, BPH1), finding that several *TAS2R* receptors are expressed in ovarian cancer, and that the transcriptional regulation of these receptor genes is variable in high-grade serous and low-grade serous ovarian cancer cell lines. The expression of *TAS2R14* and *TAS2R38* was repressed in nearly all ovarian cell lines, but *TAS2R4* and *TAS2R10* were increased compared to the control tissue samples. In prostate cancer cells, the expression of *TAS2Rs* was found to be largely decreased, except for *TAS2R38* in PC3 cells. 

In 2015, Wölfle et al. showed that the human neuroblastoma cell line SH-SY5Y is responsive to salicin, inducing neurite outgrowth in a *TAS2R16*- and Erk-mediated fashion [126].

Furthermore, the role of the *TAS2R38* receptor in pancreatic cancer was studied. It was reported that *TAS2R38* was expressed and localised intracellularly in lipid droplets of tumour cells from pancreatic cancer patients and tumour-derived cell lines. Stimulation with N-acetyl-dodecanoyl homoserine (AHL-12) and phenylthiourea (PTU) leads to the *TAS2R38*-mediated activation of key transcription factors. These results indicate the potential involvement of *TAS2R38* in pancreatic cancer progression [127].

Yamaki et al. genotyped two *TAS2R* genes, *TAS2R38* and *TAS2R46*, in Japanese cancer patients diagnosed with biliary tract cancer, hepatocellular carcinoma, pancreatic cancer, colorectal cancer and gastric cancer. They found that cancer risk is not associated with any *TAS2R46* genotypes, although they confirmed that phenylthiocarbamide (PTC) non-tasters homozygous (AVI/AVI) for *TAS2R38* were more frequent among Japanese cancer patients. Therefore, it might be considered to increase cancer risk [128].

In a cohort study of colorectal adenoma cases, researchers explored the association between colorectal adenoma risk, dietary intake and genetic variation in three *TAS2R* genes: *TAS2R38* (rs713598, rs1726866, rs10246939), *TAS2R16* (rs846672) and *TAS2R50* (rs1376251). No significant associations were observed between the *TAS2R38* PAV/PAV diplotype or *TAS2R16* (rs846672) polymorphism and the selected diet variables [129]. A meta-analysis performed by Choi et al. showed that the *TAS2R38* diplotype minimally modified the GI neoplasm risk. Further studies should be performed to verify these hypotheses and the true role of *TAS2R38* in gastrointestinal carcinogenesis [130].

A study by Tsutsumi et al. examined the effect of chemotherapy and radiotherapy on the expression of taste receptors in lingual mucosa scrapings from head and neck cancer patients [131]. The tumour types included laryngeal, hypopharyngeal, pharyngeal and maxillary cancer, among others. In 17 patients with mild/moderate stomatitis, the mRNA levels of *TAS2R5* mRNA were significantly and transiently increased after chemotherapy. Although the perception threshold of bitter taste remained unchanged, lingual mRNA levels of *TAS2R5* were significantly increased in patients who complained of phantogeusia after chemotherapy. Additionally, a genomic study on oral cancer patient cohorts demonstrated an association between a coding SNP (rs1015443) in the *TAS2R13* gene and increased susceptibility to alcohol consumption. Considering that strong alcohol consumption is a risk factor for head and neck cancer, this discovery may have an important role and should be further investigated [132]. There is no doubt that further studies are needed to investigate the chemosensory role of *TAS2Rs* in cancer. 

## 9. Conclusions

Over the past decade, bitter taste receptors have not only been found outside the mouth and throat, but their functions have been proved to go beyond just gustatory purposes. Their protective properties have been proven in both the upper and lower respiratory tracts by functioning in airway ciliated cells as a sentinel receptor for detecting invading Gram negative bacteria and triggering a critical defensive bactericidal response or mediating bronchodilatation. The polymorphism of *TAS2R* genes might be an important factor in predicting the course of chronic rhinosinusitis, defining the manner and intensity of the treatment proposed to patients. *TAS2Rs* may also play an important role in pathological states such as asthma or diabetes, creating new areas for developing new medicines using signalling pathways different from those used so far. Their presence in cancer cells may be used for monitoring treatment and possible recurrence. Indisputably, *TAS2R* receptors have a lot of information and potential that still needs to be revealed.

## Figures and Tables

**Figure 1 ijms-21-05156-f001:**
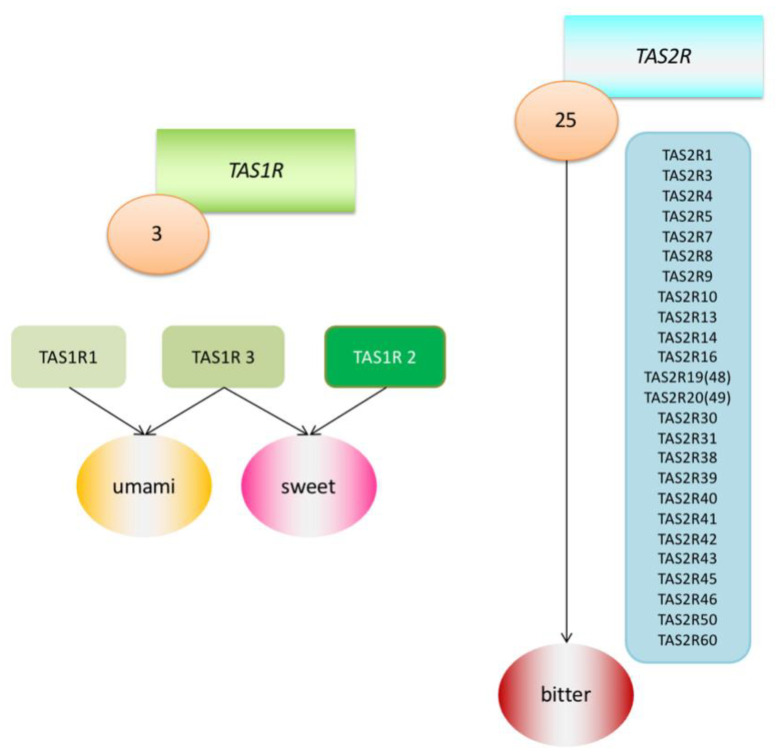
Graphic model of Tas1r (TAS1R1, TAS1R2, TAS1R3) and *TAS2R* (25 *TAS2Rs*) receptor families.

**Figure 2 ijms-21-05156-f002:**
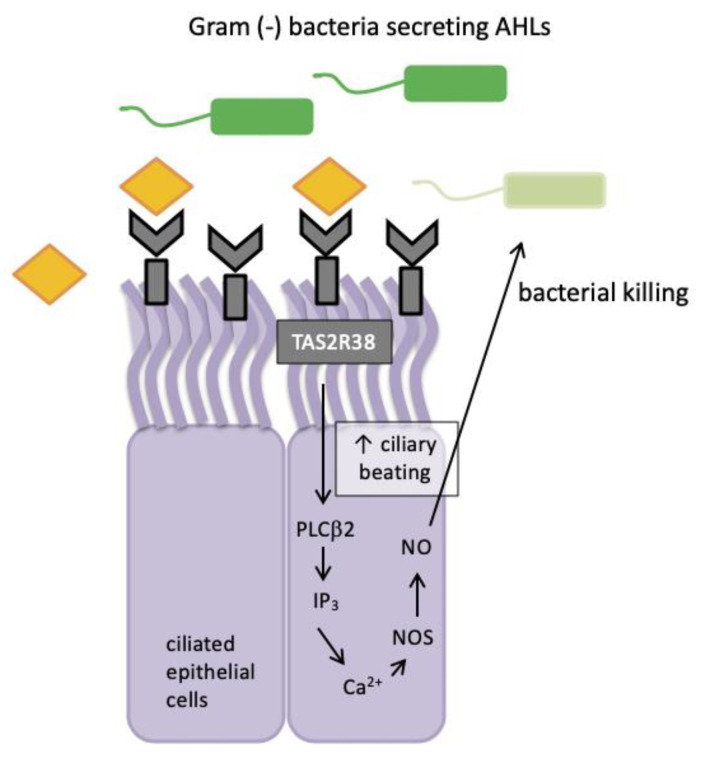
*TAS2R38* bitter taste receptor regulation in human sinonasal epithelial innate immunity. AHLs = acyl-homoserine lactones; Ca^2+^ = calcium ion; NO = nitric oxide; NOS = nitric oxide synthase; PLCβ2 = phospholipase; IP_3_ = inositol trisphosphate.

**Table 1 ijms-21-05156-t001:** Selected studies on *TAS2R* receptors in different human tissue lines.

Author	Tissue	*TAS2R*	Method	Main Outcome
**Mennella 2005** [13]	buccal mucosa cells	*TAS2R38*	qRT-PCR	Genotypes at the *TAS2R38* locus were significantly related to preferences for sucrose and sweet-tasting beverages and foods, such as cereals, in children but not in adults.
**Rozengurt 2006** [14]	cultured intestinal HuTu-80 and NCI-H716 cell lines	*TAS2R3/4/5* *TAS2R10/13/14* *TAS2R38/39* *TAS2R39* *TAS2R40/42/43/44/45/46/47* *TAS2R50* *TAS2R60*	qRT-PCR	NCI-H716 and HuTu-80 express transcripts that encode the GI peptides PYY and GIP and the precursor for glucagon/GLP-1, and its secretion may be via *TAS2R* receptors.
**Shah 2009** [15]	airway epithelium	*TAS2R4* *TAS2R38* *TAS2R43/46*	qRT-PCR	*TAS2Rs* are localised in cilia, and their signalling pathway is functional.
**Deshpande 2010** [16]	cultured airway smooth muscle (ASM)	*TAS2R10/14* *TAS2R31*	qRT-PCR	Bitter tastant induces bronchodilatory effect mediated by *TAS2Rs*.
**Le Neve 2010** [17]	duodenal cell line HuTu-80	*TAS2R7* *TAS2R14*	qRT-PCR	H.g.-12, a steroid glycoside purified from *H. gordonii* extract, activates human bitter receptors *TAS2R7* and *TAS2R14*.
**Jeon 2011** [18]	epithelial colorectal cell line Caco-2	*TAS2R38*	qRT-PCR	Phenylthiocarbamide was found to increase ATP-binding cassette B1 expression and increases its efflux activity.
**Lee 2012** [19]	sinonasal epithelial cells	*TAS2R38*	qRT-PCR	The *TAS2R38* genotype represents a defining characteristic in respiratory innate defence that contributes to the complex genetic and environmental interactions predisposing humans to upper respiratory infections.
**Grassin-Delyle 2013** [20]	lung tissue	*TAS2R5/7* *TAS2R10/14* *TAS2R38/39* *TAS2R43*	qRT-PCR	*TAS2R5, 10* and *14* are involved in the relaxation of human bronchi.
**Orsmark-Pietras 2013** [21]	WBC from children with severe asthma and WBC from healthy children	*TAS2R4/5,* *TAS2R10/13/14/19* *TAS2R20* *TAS2R31* *TAS2R45/46* *TAS2R50*	qRT-PCR	The expression of most bitter taste receptors was higher in children with severe asthma compared to the healthy controls, and these differences reached statistical significance for *TAS2R13, TAS2R14* and *TAS2R19.* The expression of all bitter taste receptors was significantly greatest in the lymphocyte population compared to monocytes.
**Foster 2013** [22]	heart tissue	*TAS2R3/4/5* *TAS2R13/14/19(48)* *TAS2R20(49)* *TAS2R30/31* *TAS2R43/45/46* *TAS2R50*	qRT-PCR	*TAS2R14* was the most expressed *TAS2R* in human heart tissue.
**Lund 2013** [23]	mesenchymal stromal cells (MSC)	*TAS2R46*	qRT-PCR	Human bone marrow expresses the *TAS2R46* receptor, and it is functional.
**Singh 2014** [24]	mammary epithelial cell lines MCF-10A, MCF-7 and MDA-MB-231	*TAS2R1/4* *TAS2R10* *TAS2R20(49)* *TAS2R38*	qRT-PCR	*TAS2R4* was the highest expressed *TAS2R* in normal mammary epithelial cells compared to breast cancer cells.
**Ekoff 2014** [25]	cord blood-derived mast cells (CBMCs) and the mast cell line HMC1.2	*TAS2R3/4/5* *TAS2R10/14/19* *TAS2R20(49)* *TAS2R46*	qRT-PCR	*TAS2Rs* mediated significant inhibition of the release of histamine and PGD2 from IgE-receptor-activated primary human mast cells. The *TAS2Rs* may mediate anti-inflammatory responses.
**Clark 2015** [26]	thyroid cells, thyrocyte line Nthy Ori 3-1	*TAS2R4* *TAS2R10* *TAS2R38* *TAS2R42/43*	qRT-PCR	*TAS2R42* is associated with differences in the circulating levels of thyroid hormones.
**Yu 2015** [27]	enteroendocrine NCI-H716 cells	*TAS2R38*	qRT-PCR	GLP-1 secretion was significantly increased by berberine incubation, which was concentration-dependently inhibited by the *TAS2R38* antibody.
**Wölfle 2015** [28]	primary keratinocytes (HPKs), keratinocyte cell line HaCaT	*TAS2R1* *TAS2R38*	qRT-PCR	*TAS2R* expression was shown throughout the epidermis, but it was weak in the basal cells.
**Wölfle 2016** [29]	placental cell line JEG-3	*TAS2R38*	qRT-PCR	The expression of fully functional *TAS2R38* receptors was found in placental tissue.
**Jaggupilli 2017** [30]	cystic fibrosis bronchial epithelial cells (CuFi-1) normal bronchial epithelial cells (NuLi-1)	*TAS2R3/4/5* *TAS2R10/13/14/19(48)* *TAS2R320(49)*	qRT-PCR	The expression of *TAS2R14* and *TAS2R20 (49)* is much higher in breast cancer cells than in normal mammary tissue.
	airway smooth muscle cells (ASM)	*TAS2R14* *TAS2R20(49)* *TAS2R50*		
	pulmonary artery smooth muscle (PASM)	*TAS2R14* *TAS2R20(49)* *TAS2R50*		
	mammary epithelial cells MCF-10A	*TAS2R14* *TAS2R20(49)*		
	breast cancer cells MDA-MB-231	*TAS2R14* *TAS2R20(49)*		
**Latorre 2016** [31]	colonic mucosal calls	*TAS2R38*	qRT-PCR	*TAS2R38* is expressed by the human colonic mucosa, where it is localised to distinct types of enteroendocrine cells (EEC), including cholecystokinin (CCK), glucagon-like peptide-1 (GLP-1) and peptide YY (PYY) cells.
**Zheng 2017** [32]	myometrial cells, hTERT-HM cells	*TAS2R1/3/4/5/7/8/9* *TAS2R10/13/14/16/19* *TAS2R20* *TAS2R30/31/38/39* *TAS2R40*	qRT-PCR	Human myometrial cells express *TAS2Rs* and the activation of the canonical *TAS2R* signalling system by bitter tastants in myometrial cells produces profound relaxation of the myometrium.
**Hariri 2017** [33]	sinonasal tissue	*TAS2R14* *TAS2R38*	qRT-PCR	Flavones activate *TAS2R14* expression in sinonasal epithelial cell cilia.
**Freud 2018** [34]	sinonasal tissue	*TAS2R4* *TAS2R14/16* *TAS2R38*	qRT-PCR	All examined *TAS2Rs* are activated by bacterial ligands.
**Shaw 2018** [35]	skin cells	*TAS2R3/4/5/9* *TAS2R13/14/16/19* *TAS2R20* *TAS2R30/31/38/39* *TAS2R40/41/42/43/45/46* *TAS2R50* *TAS2R60*	qRT-PCR	There are differences in *TAS2R* expression in the skin concerning sun exposure, sex and age.
**Grassin-Delyle 2019** [36]	lung macrophages (LM)	*TAS2R3/4/5/7/8/9* *TAS2R10/14/19* *TAS2R20* *TAS2R31/38/39* *TAS2R43/45/46*	qRT-PCR	*TAS2Rs* are expressed in human LMs, and *TAS2R* agonists inhibit LPS-induced cytokine release by LMs—a process that is not mediated by the release of IL-10.
**Cont 2019** [37]	buccal mucosa	*TAS2R38*	qRT-PCR	Infants insensitive to bitter taste (AVI/AVI) were more likely to consume the whole first complementary food meal at the first attempt, compared to sensitive ones (AVI/PAV or PAV/PAV).
**Governini 2020** [38]	semen cells, testicular tissue	*TAS2R3/4* *TAS2R14/19* *TAS2R43*	ddPCR	*TAS2R14* is the most expressed bitter receptor subtype in both testis tissue and sperm cells. In vitro capacitation significantly affects the expression and the subcellular localisation of *TAS2R* receptors in isolated spermatozoa.

GI = gastrointestinal, PYY = peptide YY, GIP = gastric inhibitory peptide, GLP-1 = glucagon-like peptide-1, PGD2 = prostaglandin D2, EEC = enteroendocrine cells, CCK = cholecystokinin, ASM = airway smooth muscle, PASM = pulmonary artery smooth muscle, LM = lung macrophages, CBMCs= cord blood-derived mast cells, MSC = mesenchymal stromal cells, WBC = white blood cells, HPKs = human primary keratinocytes.

**Table 2 ijms-21-05156-t002:** *TAS2R38* polymorphisms in selected studies in patients with chronic rhinosinusitis (CRS) and cystic fibrosis (CF).

Author	Study Group	Pathological Condition	Tissue	PAV/PAV	PAV/AVI	AVI/AVI
Adappa 2013 [70]	28	CRS	sinonasal tissue samples	1	14	13
Adappa 2014 [71]	70	CRS	sinonasal tissue samples	6	38	26
Adappa 2016 [72]	123	CRS	sinonasal tissue samples	27	96	
Dżaman 2016 [73]	20	CRS	blood	4	10	6
Gallo 2016 [77]	53	CRS	blood	8	32	13
Adappa 2016 [78]	49	CF	sinonasal tissue samples	7	24	18
Cohen 2017 [76]	28	CRS	sinonasal tissue samples	1	14	13
Cantone 2018 [74]	124	CRS	saliva blood	2	32	31

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
