# Peer review of "Clinical Role of Extraoral Bitter Taste Receptors"

_ijms, 2020, doi:10.3390/ijms21145156_

Round 1

Reviewer 1 Report

This review describes the role of bitter taste receptors expressing in extra-oral tissues. The roles are well described mainly for human tissues and derived cultured cells from the medical points of views; however, there are some sentences to be improved before publication.

L17: There are more than 50 TAS2R genes coding bitter taste receptors.

In the nomenclature, TAS2R means human bitter taste receptors whose functional genes are 25-26. Therefore, it is better to limit the description to human bitter taste receptors like “There are about 25 functional TAS2R genes in human genome, which encodes T2R bitter taste receptor proteins.”

L40: It is better to refer more recent papers especially on sour taste, otop.

L56 and Figure 1: related to L17, it is better to limit to human case. In Figure 1, what mean 100 around TAS2R?

L66: 25 T(A)2Rs

L92-93: T2R16 and T2R38 are not broadly tuned receptors.

L323: delete “Moreover”

L332: these cells ->JEG3 cell

L366-369: It is better to divide to 2 sentences.

L455: our findings ?

L468: the T2R38 role -> the role of T2R38

Author Response

Response to Reviewer 1 Comments

Dear Reviewer 1,

We would like to thank you for your discerning analysis of my manuscript.

We find all your points extremally valid and all were corrected as indicated.

Point 1: L17: There are more than 50 TAS2R genes coding bitter taste receptors.

In the nomenclature, TAS2R means human bitter taste receptors whose functional genes are 25-26. Therefore, it is better to limit the description to human bitter taste receptors like “There are about 25 functional TAS2R genes in human genome, which encodes T2R bitter taste receptor proteins.”

Response 1: We agree that the exact number of TAS2R genes is not precisely known and it is better to stay within those confirmed as ones encoding functional receptors in human. It is now: There are about 25 functional TAS2R genes coding bitter taste receptors proteins.

Point 2: L40: It is better to refer more recent papers especially on sour taste, otop.

Response 2: We added 3 recent references. Two of them refer strictly to sour taste receptors and one to all the receptors in context of neuronal transmission.

Lee H, Macpherson LJ, Parada CA, Zuker CS, Ryba NJP. Rewiring the taste system. Nature. 2017;548(7667):330-333. doi:10.1038/nature23299

Challis RC, Ma M. Sour taste finds closure in a potassium channel. Proc Natl Acad Sci U S A. 2016;113(2):246-247. doi:10.1073/pnas.1523319113

Bushman JD, Ye W, Liman ER. A proton current associated with sour taste: distribution and functional properties. FASEB J. 2015;29(7):3014-3026. doi:10.1096/fj.14-265694

Point 3: L56 and Figure 1: related to L17, it is better to limit to human case. In Figure 1, what mean 100 around TAS2R?

Response 3: The Figure 1 was corrected to enumerate only the 25 functional TAS2R genes that encode receptors in humans. Graphic part was also changed.

Point 4: L66: 25 T(A)2Rs

Response 4: This is a typo resulting from T(AS)2Rs.

Point 5: L92-93: T2R16 and T2R38 are not broadly tuned receptors.

Response 5: This is a very important point. The mistake is due to our oversight. Human TAS2R16 respond selectively to bitter ß-glucopyranosides and TAS2R38 thioamides. We have corrected this accordingly in the manuscript.

Point 6: L323: delete “Moreover”

Response 6: We agree that this sentence function better without “moreover”.

Point 7: L332: these cells ->JEG3 cell

Response 7: We added the JEG3 cells name in brackets.

 Point 8: L366-369: It is better to divide to 2 sentences.

Response 8: We agree that dividing this sentence into two makes it clearer – it was revised accordingly.

Point 9: L455: our findings?

Response 9: This typographical error was corrected accordingly.

 Point 10: L468: the T2R38 role -> the role of T2R38

Response 10: I agree that this way the sentence is better. It is changed into “The role of TAS2R38 receptor”

Reviewer 2 Report

 The paper “Clinical role of extraoral bitter taste receptors” is a high quality systematic study integrating biological aspect and clinical data. The biological implications are well articulated.

 The writing is clear, concise, well conducted and contains the appropriate references.

I suggest only two recommendations:

1)Maybe in figure 1 there is a mistake with a number 100 referred to T2R

2)In the references for some articles the DOI is reported and for others it is not, also sometimes DOI is reported with upper and sometimes lower case letters (doi) (for example lines 525, 543, 545, 548 and many others)

Author Response

Response to Reviewer 2 Comments

Dear Reviewer 2,

We would like to thank you for the analysis of my manuscript and for your valid points.

Point 1: Maybe in figure 1 there is a mistake with a number 100 referred to T2R

Response 1: Yes, there is an error on the graphic part of the Figure. It should have number 25, not 100 as this represents the number of functional receptor types rather than genes encoding potential receptor proteins. It was corrected accordingly.

 Point 2: In the references for some articles the DOI is reported and for others it is not, also sometimes DOI is reported with upper and sometimes lower case letters (doi) (for example lines 525, 543, 545, 548 and many others)

Response 2: As IJMS does not require doi numbers or links in references we deleted them. Other minor discrepancies in the format of some citations may have been caused by automated import using Mendeley – all these minor mistakes were corrected and the reference list is now uniformly formatted.

Reviewer 3 Report

Jeruzal-Swiatecka et al provide an up-to-date and comprehensive review on the pathophysiological roles of extraoral bitter taste receptors.  It is appropriate that authors focus on reviewing human studies as several reviews on the same topic including animal data have recently been published in other journals.  Overall, the review is objective, balanced, well-organized and well-written.  However, there are concerns need to be addressed.

  1. There is a confusion on the number of human bitter taste receptors. On page 2 line 56, authors state that “there are more than 50 TAS2R genes that code bitter taste receptors.”  However, figure 1 caption indicates only 25 TA[S]2Rs.  Also what does 100 in Figure 1 (right) refer to?  
  2. Authors use interchangeably T2R and TAS2R throughout the review. Authors should stick to one nomenclature, preferably TAS2R.  The same concern applies to T1R and TAS1R.
  3. Two references (PMID: 31963712; PMID: 28559440) on TAS2Rs in the reproductive system are missing. They should be included in Table 1 and discussed.

Author Response

Response to Reviewer 3 Comments

Dear Reviewer 3,

We would like to thank you for the analysis of my manuscript and for your valid points. All were corrected as indicated.

Point 1: There is a confusion on the number of human bitter taste receptors. On page 2 line 56, authors state that “there are more than 50 TAS2R genes that code bitter taste receptors.”  However, figure 1 caption indicates only 25 TA[S]2Rs.  Also what does 100 in Figure 1 (right) refer to?  

Response 1:

This is a very important point. The exact number of TAS2R genes is not precisely known and it is better to stay within those confirmed as ones encoding functional receptors in human. That is why we decided to refer only to those 25 genes and their protein products. Lines 17 and 55 have been changed due to this correction. There was an error on the graphic part of Figure 1 as its number related to the number of genes rather than those encoding functional receptors. Therefore, it was corrected to enumerate only the 25 functional TAS2R genes that encode receptors in humans and this number was cited throughout the manuscript.

Point 2: Authors use interchangeably T2R and TAS2R throughout the review. Authors should stick to one nomenclature, preferably TAS2R.  The same concern applies to T1R and TAS1R.

Response 2: We intended to distinguish this way those moments when we refer to gene and when to the receptor itself. While using TAS2R we were referring to gene and while using T2R38/16/14 we were referring to receptor. It is a valid point that we should stick to one nomenclature, so we changed it all to TAS2R while referring to receptor as it is more genetically appropriate. It was also changed in graphic and description of Figure 1, Figure 2 and Table 1.

Point 3: Two references (PMID: 31963712; PMID: 28559440) on TAS2Rs in the reproductive system are missing. They should be included in Table 1 and discussed.

Response 3: We have cited the relevant articles in the manuscript. We included them in Table 1 and discussed them in section: 5. Bitter taste receptors in other systems and diseases.
